# Enrofloxacin, Effective Treatment of *Pseudomonas aeruginosa* and *Enterococcus faecalis* Infection in *Oreochromis niloticus*

**DOI:** 10.3390/microorganisms12050901

**Published:** 2024-04-30

**Authors:** Ibrahim Aboyadak, Nadia Gabr Ali

**Affiliations:** National Institute of Oceanography and Fisheries, NIOF, Cairo 4262110, Egypt; i.aboyadak@gmail.com

**Keywords:** Nile tilapia, bacterial infection, LD_50_, pathogenicity, histopathology, treatment

## Abstract

Enrofloxacin is a broad-spectrum synthetic antimicrobial drug widely used in veterinary medicine. The present study aimed to determine the effective enrofloxacin dose for treating *Pseudomonas aeruginosa* and *Enterococcus faecalis* infection in *Oreochromis niloticus. P. aeruginosa* and *E. faecalis* isolates were verified using selective differential media and biochemically using the Vitek 2 test. Bacterial isolates were virulent for *O. niloticus* with LD_50_ equal to 2.03 × 10^6^ and 2.22 × 10^7^ CFU fish^−1^ for *P. aeruginosa* and *E. faecalis*, respectively. Infected fish suffered from decreased feed intake followed by off-food, tail erosion, darkening of the external body surface, exophthalmia, ascites, and loss of escape reflex. Internally, congested hemorrhagic hepatopancreas with engorged distended gall bladder were dominant. The posterior kidney was congested with enlarged spleen, and empty elementary tract. Pathologically, severe degenerative changes were dominant in the hepatopancreas, posterior kidney, spleen, stomach, and gills of infected fish. Antimicrobial sensitivity test indicated the high susceptibility of *P. aeruginosa* and *E. faecalis* to enrofloxacin with MIC estimated at 1 and 0.0625 µg/mL, respectively. Enrofloxacin effectively protected *O. niloticus* against *E. faecalis* and *P. aeruginosa* infection when used with medicated feed at doses of 10 and 20 mg kg^−1^ body weight.

## 1. Introduction

In Egypt, *Oreochromis niloticus* is the first economically important fish; cultured tilapia production exceeded 1.1 million tons in 2020 [1]. Summer mortality syndrome represents the most severe challenge facing tilapia culture in Egypt over the past few years [2]. Disease outbreaks hit most tilapia farms in Egypt [3]; *A. veronii, A. hydrophila, A. caviae, A*. *sobria*, *Pseudomonas* sp., and *Streptococcus* sp. isolated from the affected farms [4].

Bacterial infections are the most abundant diseases affecting cultured fish [5]. Globally, fish diseases are estimated to contribute to more than 30% of the overall production loss, and bacterial diseases represent a serious challenge for tilapia culture worldwide [6]. Gram-negative bacteria related to the genus *Aeromonas*, *Pseudomonas*, *Vibrio*, and *Flavobacterium* are responsible for high mortality rates and severe economic losses in cultured fish and shrimp [7,8]. *Pseudomonas aeruginosa* is one of the most virulent Gram-negative pathogens affecting many cultured fresh and marine fish species, including *O. niloticus*, *Clarias gariepinus*, *Dicentrarchus labrax, Oncorhynchus mykiss* and *Sparus aurata* [9,10,11,12]. *E. faecalis* is a newly emerged Gram-positive fish pathogen responsible for high mortality rates in the aquaculture of various fish species worldwide. *E. faecalis* infections were recorded in cultured *O. niloticus*, *Barbonymus gonionotus*, *Cyprinus carpio*, *Oncorhynchus* sp., *Pterogymnus laniarius*, *Scophthalmus maximus*, stinging catfish and walking catfish [13,14,15,16,17].

Enrofloxacin is a synthetic antibacterial drug related to fluoroquinolones and remains extensively used in veterinary medicine [18]. Enrofloxacin acts by inhibiting the DNA gyrase enzyme (topoisomerase II); DNA gyrase is responsible for the standard coiling of DNA within the nucleus [19]. Enrofloxacin has a potent broad-spectrum bactericidal activity at a relatively minute concentration due to its low MIC against many Gram-negative animal pathogens; it also has a high post-antibacterial effect [20]. Enrofloxacin exhibits a favorable pharmacokinetic profile [21] and is effective against many bacterial fish pathogens, including *Aeromonas*, *Pseudomonas*, *Vibrio*, and *Renibacterium salmoninarum* [22].

This study aimed to determine the efficacy and proper dose of enrofloxacin in treating *P. aeruginosa* and *E. faecalis* as examples of Gram-negative and Gram-positive bacterial infections in *O. niloticus.*

## 2. Materials and Methods

### 2.1. Experimental Fish

A total number of 500 *O. niloticus* fingerlings were used in the infectivity test and the treatment trial. Fishes ranged between 10–13 cm in total length and 30–40 g in body weight. Fish were purchased from a private farm located at Trombat 7 district, Riyadh city, Kafrelshiekh Provence. Fish were transported to the wet laboratory, Baltim station, National Institute of Oceanography and Fisheries, under optimum transportation conditions mentioned by Ali [23]. During transportation, the water temperature decreased by 5 °C to slow-down fish activity. Twenty-five mg L^−1^ of tricaine methane sulfonate was used for fish tranquillization [24]. Continuous aeration was maintained during the transportation process using pure oxygen cylinders. After transportation, fish were maintained off-food for 24 h and observed during acclimatization. Five randomly selected fishes were dissected to detect any parasitic infestation and another five fish were subjected to microbiological examination, all the examined fish were free from fish diseases.

After the end of the study, the remaining fish were euthanized using 500 mg L^−1^ of tricaine methane sulfonate (Syncaine^®^), Syndel, Washington, DC, USA. Fish were left in the anesthetic solution till complete cessation of opercular movement and then burned.

### 2.2. Enrofloxacin

Enrofloxacin base (99%) CAS number: 93106-60-6, Xi’an SENYI New Material Technology Co., Ltd., Shannxi, China.

### 2.3. Infectivity Test

The infectivity test was performed to determine the virulence of bacterial isolates against *O. niloticus* (to satisfy Koch postulates) and to calculate lethal dose fifty (LD_50_).

#### 2.3.1. Bacterial Isolates

*P. aeruginosa* and *E. faecalis* were previously isolated from a diseased *O. niloticus* farm during a summer mortality outbreak.

#### 2.3.2. Verification of Bacterial Isolates

Bacterial isolates were preliminarily identified on the selective media, *Pseudomonas* selective agar supplemented with cephalothin, fucidin, and cetrimide for *P. aeruginosa* and M-Enterococcus agar base media for *E. faecalis*. After that, isolates were reconfirmed using the Vitek 2 automatic biochemical identification system following the method described by Ali [25]. One bacterial colony (from a fresh bacterial culture) was suspended in 5 mL of 0.5% sodium chloride solution; after that was adjusted to 0.6 McFarland standards. Identification cards were inoculated with bacterial suspensions in the Vitek 2 system, and the biochemical profile was recorded.

#### 2.3.3. Bacterial Inoculum Preparation for the Infectivity Test

A single bacterial colony was picked up from the selective agar and then inoculated on brain heart infusion broth, and incubated at 35 °C for 12 h. Bacterial growth was harvested by centrifugation at 5000 rpm for 3 min. The bacterial pellet was suspended in 0.1% peptone water and adjusted with a spectrophotometer to 0.451 absorbances at 600 nm (equivalent to the second McFarland standard 6 × 10^8^ CFU mL^−1^). One ml of sterile phosphate buffer saline was added to five ml of bacterial suspension to achieve a final concentration of 5 × 10^8^ CFU mL^−1^. Tenfold serial dilutions were prepared four consecutive times to obtain the following concentrations (5 × 10^7^, 5 × 10^6^, 5 × 10^5^, and 5 × 10^4^) CFU mL^−1^.

#### 2.3.4. Experimental Design

Two hundred and eighty-eight fish were randomly divided into twelve groups as in Table 1; each group consists of 24 fish in three triplicates (8 fish per replicate). Fish in groups (1–5) were intraperitoneally inoculated with 0.2 mL of *P. aeruginosa* bacterial suspension containing (5 × 10^4^, 5 × 10^5^, 5 × 10^6^, 5 × 10^7^ and 5 × 10^8^) CFU mL^−1^ equivalent to (10^4^, 10^5^, 10^6^, 10^7^ and 10^8^) CFU fish^−1^. Fish in the last group were inoculated with 0.9% saline and served as a control negative. The same experimental design was performed for groups (6–10) using *E. faecalis* bacterial suspension. Each replicate was maintained in a 100 L glass aquarium; water temperature was thermostatically maintained at 28 ± 1 °C, and aquaria water was changed at a continuous rate (5 Liter per hour). Feeding was restricted for 24 h before the challenge and then resumed 12 h after infection. All fish groups were observed for seven days to record the clinical signs, postmortem lesions, and mortality rates. Dead fish were considered only after the re-isolation of challenging bacteria, and LD_50_ was calculated according to Reed and Muench [26].

### 2.4. Clinical Picture

Fish were observed daily throughout the experimental period to record any abnormal signs and behavioral changes, as described by Austin and Austin [27]. Dead fish were immediately dissected under aseptic conditions for bacterial re-isolation. After that, the gross internal lesions were recorded, and tissues were sampled for the histopathological examination.

### 2.5. Histopathological Investigation

The histopathological examination was performed according to Suvarna et al. [28]. Small pieces from the hepatopancreas, gills, spleen, stomach, and posterior kidney of moribund fish were fixed in 10% buffered formalin. Fixed tissues were dehydrated in ascending-grade ethyl alcohol and then cleared in xylene. Cleared samples were impeded in soft then hard paraffin wax and sectioned to 5 µm thickness using Leica RM2235 microtome (Lecia, Germany). Thin sections were mounted over labeled glass slides and finally stained with hematoxylin and eosin. Stained slides were examined and photographed using a microscope equipped with a digital camera.

### 2.6. Antimicrobial Susceptibility Tests

#### 2.6.1. Agar Disc Diffusion Test

Susceptibility of *P. aeruginosa* and *E. faecalis* to enrofloxacin was assayed. Overnight-seeded broth was adjusted to 1.5 × 10^8^ CFU mL^−1^, and then 2 mL was spread on the Mueller-Hinton agar (Oxoid, UK) plate surface with a rotation movement. The plate was allowed to stand in an inverted position on the refrigerator for 20 min to absorb the excess moisture. The sensitivity disc (ENR 5 µg), Oxoid, UK, was gently fixed into the agar surface using sterile forceps. The agar plate was incubated at 35 °C for 24 h, and *Escherichia coli* ATCC 25,922 was used as a control. The inhibition zone was measured to the nearest mm using a digital caliper and interpreted as susceptible (21 mm or more), intermediate susceptible (16–20 mm), and resistant (less than 15 mm) according to breakpoints mentioned by CLSI [29].

#### 2.6.2. Broth Dilution Test

The minimum inhibitory concentration (MIC) of enrofloxacin was determined for the tested isolates using the broth dilution test, as indicated by Ali et al. [23]. Briefly, 256 µL of Enrofloxacin 10% was added to 1744 µL sterile distilled water. Double-fold serial dilution was performed 15 successive times. The overnight cultured tryptic soy broth was adjusted to 0.5 McFarland standard (absorbance of 0.063 at 600 nm). One ml of TSB was added to 199 mL of sterile Mueller-Hinton broth. After that, tetrazolium chloride (20 mg) was added as an indicator for bacterial growth. Each screw-capped test tube was loaded with 4.9 mL of seeded Mueller-Hinton broth (a final volume of 5 mL). After that, 100 μL from previously prepared enrofloxacin standard solution was added to the corresponding test tubes to achieve a final concentration of 265, 128, 64, 32, 16, 8, 4, 2, 1, 0.5, 0.25, 0.125, 0.0625, 0.03125 and 0.015625 μg mL^−1^, respectively. The last tube was left antibiotic-free as a control; tubes were incubated at 35 °C for 24 h. MIC was determined as the lowest antibiotic concentration, preventing bacterial growth. Red-colored broth indicated bacterial growth. Results interpreted as susceptible when MIC equals (1 µg/mL or less), intermediately susceptible (2–4 µg/mL), and resistant (more than 4 µg/mL) as mentioned by CLSI [29].

#### 2.6.3. Protective Effect of Enrofloxacin against *P. aeruginosa* and *E. faecalis* Challenge

The protective effect of enrofloxacin against *P. aeruginosa* and *E. faecalis* infection in *O. niloticus* was assayed as the following: Two hundred and ten fingerlings were randomly divided into eight groups, as shown in Table 2. Groups (11 and 12 and +ve control) were experimentally infected through intraperitoneal injection with *P. aeruginosa* 2.03 × 10^6^ CFU fish^−1^, while groups (13 & 14 & +ve control) received *E. faecalis* 2.22 × 10^7^ CFU fish^−1^, control -ve groups received 0.2 mL of normal saline. Groups (11 and 13) received medicated feed containing enrofloxacin 10 mg kg^−1^ body weight equivalent to 340 mg kg^−1^ fish ration at (3%) feeding rate of fish weight. Groups (12 and 14) received 20 mg kg^−1^ (equivalent to 680 mg kg^−1^ ration). Enrofloxacin powder was mixed with 10 mL of fish oil, and the mixture was evenly distributed to one kg of fish feed. Medicated feed was left for one day at room temperature to absorb the drug and then preserved at 8 °C. The experimental infection was performed after consumption of the medicated feed by all the treated groups; treatment continued for seven successive days; the mortality rate was daily recorded for ten days.

## 3. Results

### 3.1. Verification of Bacterial Isolates

*P. aeruginosa* was grown as yellowish-green colonies against a greenish background on *Pseudomonas* selective agar, while *E. faecalis* raised as dark red colonies on M-*Enterococcus* agar base media Figure 1a,b. Tested isolates were confirmed as *P. aeruginosa* and *E. faecalis* with 99% probability using the VITEK 2 automated biochemical identification system. The biochemical characteristics of both pathogens are recorded in Table 3.

### 3.2. Antimicrobial Susceptibility

The agar disc diffusion test indicated that both bacterial isolates were susceptible to enrofloxacin with 21.5- and 56.5-mm zone diameters for *P. aeruginosa* and *E. faecalis*, as represented in Figure 1c,d. Broth dilution test confirmed the high susceptibility of *P. aeruginosa* and *E. faecalis* to enrofloxacin with MIC equals 1 and 0.0625 µg/mL, respectively Figure 1e,f.

### 3.3. Infectivity Test Result

The LD_50_ of *P. aeruginosa* in challenged Nile tilapia was 2.03 × 10^6^ CFU. Fish^−1^, and it was 2.22 × 10^7^ CFU. Fish^−1^ for *E. faecalis*, the mortality rate is demonstrated in Table 1.

Fish number in each group = 24.

### 3.4. Clinical Picture

Infected fish with *P. aeruginosa* or *E. faecalis* showed similar clinical signs in which fish suffered from decreased feed intake followed by off-food, with disease progression tail erosions and darkening of the external body surface taking place. Some infected fish showed exophthalmia and ascites, Figure 2a,b; fish swam near or at the water surface and lost escape reflex shortly before death.

Pale hepatopancreas tinged with petechial hemorrhages or even large hemorrhagic spots was the most prominent gross internal finding observed during the dissection of the infected fish. Enlarged distended gall bladder, congested posterior kidney, enlarged spleen, and empty intestine were also reported, as represented in Figure 2c–f.

### 3.5. Histopathological Examination

Infected *O. niloticus* with *P. aeruginosa* or *E. faecalis* showed severe degenerative changes in all tissue samples. Hepatopancreas of diseased fish showed diffused hepatocellular vacuolation, severe inflammation, mononuclear inflammatory cell infiltration, and the presence of necrotic foci with appendant melanomacrophage centers as represented in Figure 3a,b. The posterior kidney was also severely affected; renal corpuscles showed shanked glomeruli with dilated Bowmans’s space, the presence of interstitial hemorrhage, mononuclear cell infiltration, degenerated proximal and distal convoluted tubules, detached tubular epithelium, hyaline droplet degeneration, and tubular obliteration as observed in Figure 3c,d. The affected fish spleen demonstrated diffused lymphocytic proliferation clusters, cuboidal-shaped endothelial cells, and melanomacrophage centers Figure 4a,b. The stomach of experimentally infected fish showed destruction and detachment of mucosal lining, coagulative necrosis of some gastric glands with abundant lymphocytic infiltration in the lumen of gastric folds Figure 4c,d. The gill tissue of affected fish showed degeneration and fusion of secondary gill lamellae with sloughing of necrotic cells and epithelial lifting Figure 4e,f.

### 3.6. Result of the Treatment Trial

Enrofloxacin showed an excellent protective effect for the challenged *O. niloticus* against *P. aeruginosa* and *E. faecalis* infection. Enrofloxacin at a dose of 10 mg kg^−1^ completely protected the challenged fish against *E. faecalis* infection by decreasing the mortality rate from 54.16 in the infected non-treated group to 0%, while *P. aeruginosa* infection required a much higher dose (20 mg kg^−1^) to reduce mortality from 66.7 to 8.3 %, as represented in Table 2.

## 4. Discussion

Bacterial fish diseases are responsible for a huge annual loss estimated at USD 6 billion in 2014 [30]; this figure has increased to 9.58 in 2020 [31]. *P. aeruginosa* and *E. faecalis* are among the most common bacterial pathogens affecting cultured fishes [10,32]; the present study aimed to treat such serious infections using an effective antimicrobial drug such as enrofloxacin.

Selective media is a preliminary procedure used in microbial identification [33]. In the present study, *P. aeruginosa* grew as greenish colonies on *Pseudomonas* selective agar due to the secretion of pyocyanin pigment; other bacteria growth was inhibited by CFC supplement [34]. On the other hand, *E. faecalis* raised as dark red colonies on M-*Enterococcus* agar due to the uptake of triphenyl tetrazolium chloride and sodium azide preventing the growth of other microorganisms [35].

*P. aeruginosa* and *E. faecalis* were further verified by their specific biochemical profile with 99% probability using the Vitek 2^®^ automatic microbial biochemical identification system. Vitek 2^®^ system is among the most recent reliable techniques for identifying pathogenic bacteria [36]. Vitek 2^®^ system is the gold standard for *P. aeruginosa* identification with 100% accuracy, as described by Moehario et al. [37].

In the present work, LD_50_ of *E. faecalis* was 2.22 × 10^7^ CFU fish^−1^, so it is less virulent when compared to *P. aeruginosa* (2.03 × 10^6^ CFU fish^−1^); this could be due to many potent virulence factors *P. aeruginosa* has.

Rizkiantino et al. [38] found that the LD50 of *E. faecalis* in tilapia was 0.79 × 10^8^ CFU mL^−1^ which was slightly higher than that reported in the present work; this variation could be attributed to the difference in challenged fish size as well as the diversity of used strain. The calculated LD50 of *P. aeruginosa* was nearly like that reported by Thomas et al. [39] in tilapia which was 4.5 × 10^6^ CFU/fish.

Pyocyanin is the major virulence factor responsible for *P. aeruginosa’s* pathogenicity [40]. Dead fish showed the characteristic clinical and postmortem lesions of *Pseudomonas* septicemia, including exophthalmia, ascites, and hemorrhages over the external body surface, with congested hepatopancreas and posterior kidney. Similar results were observed by [10,14]. Furthermore, histopathological examination indicated the presence of congestion, inflammation, and degeneration of the hepatopancreas, spleen, and posterior kidney; Refs. [23,41,42] reported similar results. Virulent *P. aeruginosa* induced high mortality rate, serious clinical signs, and postmortem lesions with severe pathological tissue changes because of virulence factors that the pathogen has such as pyocyanin. Pyocyanin is essential for *Pseudomonas* pathogenicity; it has toxic effects responsible for cellular death and interferes with many cellular functions by inducing oxidative stress, altering the expression and release of many cytokines [43,44,45]. Outer membrane porin F, biofilm formation, exotoxin A, adhesins, and tissue-digesting enzymes as proteases are also responsible for the virulence of *P. aeruginosa* [46,47,48,49].

Enterococci are normal inhabitant gut microbes of human and other animal species [50]. Enterococci emerged as a potential bacterial pathogen inducing severe localized or septicemic life-threatening infections in humans and animals [51]. Infected *O. niloticus* expressed the general signs of septicemia, hemorrhagic batches on the skin, scale desquamation, and tail erosions. Internally, enlarged hemorrhagic hepatopancreas with distended gall bladder and congested spleen were clear on dissected fish. Refs. [52,53] recorded the same clinical signs from infected fish. The pathological changes involved the stomach and gills with impairment of their normal physiological functions; Elgohary et al. [16] and Abdelsalam et al. [54] reported similar descriptions. *E. faecalis* expresses many virulence factors directly responsible for disease progression in the affected animal. Enterococcal surface protein, surface aggregating protein, and large surface protein are responsible for biofilm formation, which subsequently helps in cell adherence, colonization, and evasion of the host immune system. *E. faecalis* produces extracellular metalloprotease (gelatinase) that hydrolyzes gelatin, collagen, and hemoglobin; it also produces serine protease and cytolysin A [55,56,57], these enzymes are directly responsible for disease pathogenesis.

Enrofloxacin is the most widely used fluoroquinolone in veterinary medicine; it has a broad spectrum of activity against many Gram-positive and Gram-negative bacterial pathogens affecting animals and fish [58,59]. The result of antimicrobial sensitivity tests indicated high susceptibility of *P. aeruginosa* and *E. faecalis* to enrofloxacin.

In harmony with the present research findings [10,60], they reported that 100% of *P. aeruginosa* isolates retrieved from diseased *O. niloticus* were highly sensitive to ciprofloxacin. Also, Anifowose et al. [61] found that most of the *E. faecalis* isolates recovered from *Clarias gariepinus* Juveniles were sensitive to enrofloxacin.

The treatment trial showed remarkable efficacy in protecting *O. niloticus* against challenged bacterial pathogens. Enrofloxacin effectively protects the challenged fish against *E. faecalis* at a dose of 10 mg kg^−1^, but the double dose was protective against *P. aeruginosa*. The difference in therapeutic dose could be due to the difference in MIC of both pathogens. *E. faecalis* was more susceptible to enrofloxacin than *P. aeruginosa* by four folds (MIC was 0.0625 and 1 µg mL^−1^), respectively. Enrofloxacin is an effective antibacterial agent when administrated with fish feed; it has excellent activity against sensitive fish pathogens. Moreover, it is a non-water-soluble powder, so the given dose is almost delivered to fish even if the feed remains for some time in the water.

In recent research, enrofloxacin is still used in many regions in the world for prophylaxis and treatment of cultured fish diseases; Amable et al. [62] used the subtherapeutic doses of enrofloxacin as a growth promotor and prophylactic for *Piaractus mesopotamicus* fish (the most cultured fish in Argentina), the drug was administrated in feed twice daily for 120 days. No significant difference was observed in drug resistance between the treated and control groups in the intestinal microbiota up to 90 days of the feeding trial. The antibiotic residues in meat samples showed no differences between controls and treatment. Concha et al. [63] reported that quinolones are still used in Chilean salmon farming and are currently approved for use in this industry. Among the 65 bacterial isolates from fish farms, only 4.6% showed resistance to enrofloxacin.

Oxytetracycline, oxolinic acid, flumequine, sarafloxacin, enrofloxacin, amoxicillin, erythromycin, sulfadimethoxine, ormetoprim, and florfenicol are the most used antibiotics in aquaculture worldwide [64]. Fluoroquinolones are the most common quinolones used in veterinary medicine; they are the most used class of antibiotics in aquaculture worldwide [65,66]. The residual limit of enrofloxacin is 30 μg/kg in the United States and the European Union. FAO and the WHO stipulate the allowable daily intake (ADI) of ENR as 2 μg/kg [67]. The withdrawal time of enrofloxacin should be considered before use in fish treatment; some studies estimated it at 45 days [68]. Ferri et al. [69] reported that the acceptable maximum residue limit (MRL) of enrofloxacin in finfish is 100 µg/kg. For the sustainability of the accelerated aquaculture growth as an important source of animal protein, this growth was accompanied by increased relay on antimicrobials to maintain fish health and fight diseases so, there is an urgent need for stewardship on antimicrobials use and monitoring the withdrawal time and drug residues [68].

## 5. Conclusions

*P. aeruginosa* and *E. faecalis* were highly pathogenic for *O. niloticus*, experimental infection-induced typical disease signs, high mortality rate, and severe pathological lesions. Enrofloxacin effectively protected *O. niloticus* against susceptible *P. aeruginosa* and *E. faecalis* infection when used with medicated feed at doses of 20 and 10 mg kg^−1^ body weight, respectively.

## Figures and Tables

**Figure 1 microorganisms-12-00901-f001:**
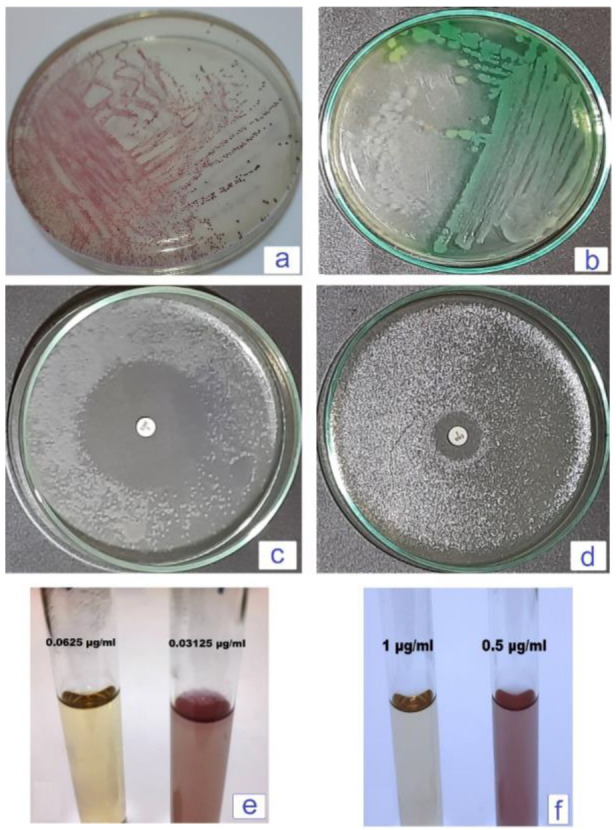
(**a**) Dark red characteristic colonies of *E. faecalis* on Enterococcus agar base media. (**b**) Yellowish green characteristic colonies of *P. aeruginosa* against a greenish background due to pyocyanin secretion on *Pseudomonas* selective. (**c**) Wide inhibition zone induced by enrofloxacin (5 µg) disc indicated high sensitivity of *E. faecalis* to tested antibacterial. (**d**) Inhibition zone induced by enrofloxacin (5 µg) disc indicated sensitivity of *P. aeruginosa* to tested antibacterial. (**e**) MIC of enrofloxacin (0.0625 µg mL^−1^) completely inhibits *E. faecalis* growth while the left tube that showed bacterial growth. (**f**) MIC of enrofloxacin is 1 µg mL^−1^ which completely inhibits *P. aeruginosa* growth in contrast with the left tube sowed bacterial growth.

**Figure 2 microorganisms-12-00901-f002:**
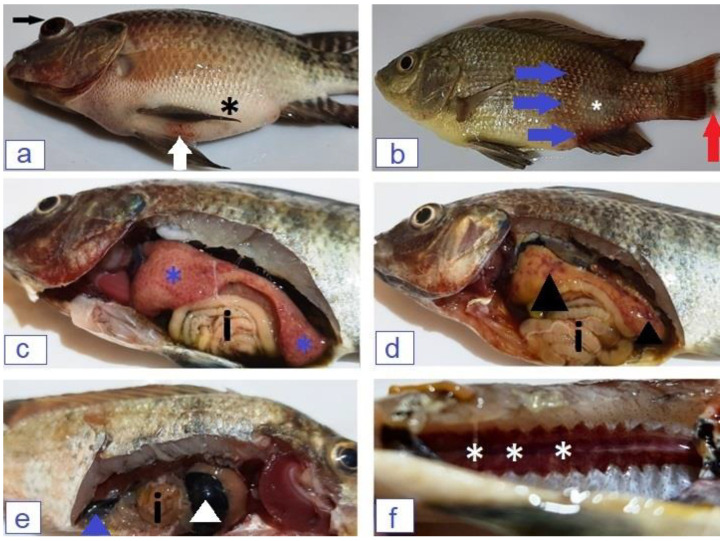
(**a**) *Oreochromis niloticus* infected with *P. aeruginosa* showed exophthalmia (black arrow), ascites (black Asterix), and the presence of hemorrhagic spots on the ventral abdominal wall (white arrow). (**b**) *O. niloticus* infected with *E. faecalis* showed hemorrhagic batches on the peduncle region (blue arrow), scale desquamation (white Asterix), and tail erosions (red arrow). (**c**) *P. aeruginosa* infected fish showed enlarged pale hepatopancreas with diffused petechial hemorrhages (blue Asterix) and empty intestine (i). (**d**,**e**) *E. faecalis* infected fish with pale enlarged hepatopancreas with the presence of hemorrhagic areas (black arrowhead), enlarged, distended gall bladder (white arrowhead), enlarged congested spleen (blue arrowhead) and empty intestine (i). (**f**) Congested posterior kidney of *P. aeruginosa*-infected fish (white Asterix).

**Figure 3 microorganisms-12-00901-f003:**
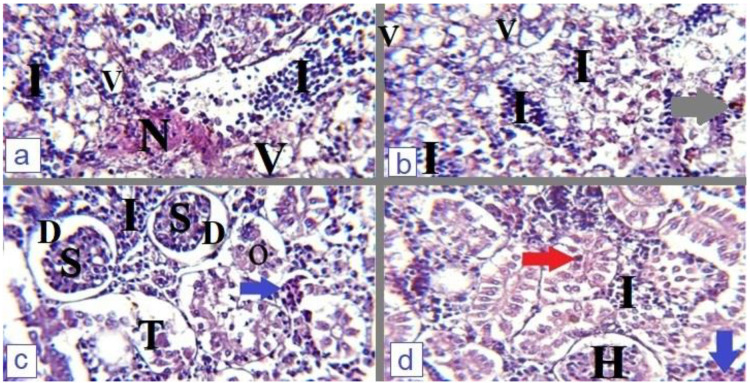
(**a**) Hepatopancreas of *O. niloticus* experimentally infected with *P. aeruginosa* showed diffused hepatocellular vacuolation (V), mononuclear inflammatory cells infiltration (I) and presence of necrotic area (N), H&E, X = 400. (**b**) Hepatopancreas of *O. niloticus* experimentally infected with *S. faecalis* showed diffused hepatocellular vacuolation (V), diffused mononuclear inflammatory cells infiltration (I) with appendant melanomacrophage centers activation (grey arrow), H&E, X = 400. (**c**) Posterior kidney of *P. aeruginosa* infected fish showed shanked glomeruli (S), dilated Bowmans’s space (D), presence of interstitial hemorrhage (blue arrow), mononuclear cell infiltration (I), degenerated proximal convoluted tubule with tubular obliteration (O) and degenerated distal convoluted tubules with detached tubular epithelium (T), H&E, X = 400. (**d**) Posterior kidney of *S. faecalis* infected fish showed hypertrophied glomeruli (H) with narrow Bowmans’s space, mononuclear cell infiltration (I), interstitial hemorrhage (blue arrow), and hyaline droplet degeneration (red arrow), H&E, X = 400.

**Figure 4 microorganisms-12-00901-f004:**
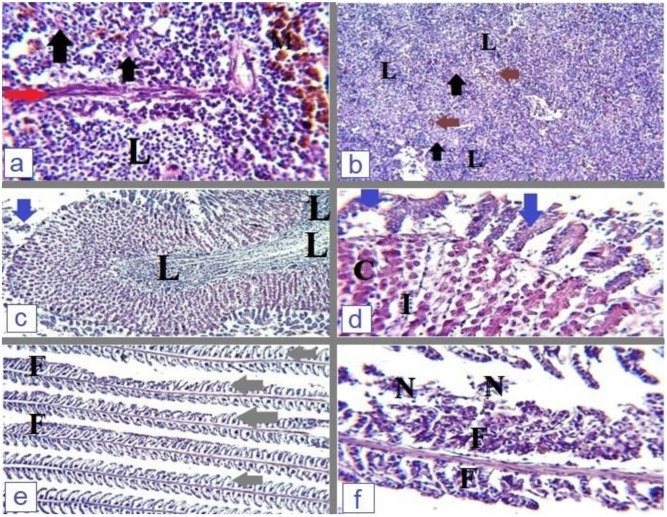
(**a**,**b**) Spleen of *S. faecalis* infected fish demonstrated diffused clusters of lymphocytic proliferation (L), cuboidal-shaped endothelial cells (black arrow), Splenic capsule-trabecula systems (red arrow) and melanomacrophage centers (M and brown arrow), H & E, X = 400 (**a**) & 100 (**b**). (**c**,**d**) Stomach of *S. faecalis* experimentally infected fish showed destruction and detachment of mucosal lining (blue arrow), coagulative necrosis of some gastric glands (C) with abundant lymphocytic infiltration between the gastric glands (I) and in the lumen of gastric folds (L), H & E, X = 100 (**c**) and 400 (**d**). (**e**,**f**) Gills of *P. aeruginosa* infected fish with degeneration and fusion of secondary gill lamellae (F) with sloughing of necrotic cells (N), and epithelial lifting (grey arrow), H & E, X = 100 (**a**) and 400 (**b**).

**Table 1 microorganisms-12-00901-t001:** Experimental design and mortality rate of *O. niloticus* fingerlings challenged with *P. aeruginosa* and *E. faecalis*.

Pathogen	*P. aeruginosa*			*E. faecalis*		
Inoculum CFU/Fish	Group No.	Dead Fish No.	Mortality %	Group No.	Dead Fish No.	Mortality %
10^4^	1	0	0	6	0	0
10^5^	2	4	1.67	7	1	4.17
10^6^	3	10	41.67	8	5	20.34
10^7^	4	17	70.84	9	8	33.34
10^8^	5	22	91.67	10	15	62.5
Normal saline	Control	0	0	Control	0	0

**Table 2 microorganisms-12-00901-t002:** Protective effect of enrofloxacin for *O. niloticus* fingerlings challenged with *P. aeruginosa* and *E. faecalis*.

*P. aeruginosa* (2.03 × 10^6^ CFU. Fish^−1^)				*E. faecalis* (2.22 × 10^7^ CFU. Fish^−1^)			
Group	Enrofloxacin Dose	Dead Fish No.	Mortality %	Group	Enrofloxacin Dose	Dead Fish No.	Mortality %
11	10 mg kg^−1^	4	16.67	13	10 mg kg^−1^	0	0
12	20 mg kg^−1^	2	8.3	14	20 mg kg^−1^	1	4.16
Control +ve	0	16	66.7	Control +ve	0	13	54.16
Control -ve	0	0	0	Control -ve	0	0	0

Fish number in each group = 24, Control +ve: Infected non-treated, Control -ve: non-infected non-treated.

**Table 3 microorganisms-12-00901-t003:** The biochemical characteristics of *P. aeruginosa* and *E. faecalis*.

Vitek Gram-Negative Identification Card.			Vitek Gram-Positive Identification Card		
Biochemical Reactions	Abbreviation	*P. aeruginosa*	Biochemical Reactions	Abbreviation	*E. faecalis*
Ala-Phe-Pro-Arylamidase	APPA	-	D-Amygdalin	AMY	+
Adonitol	ADO	-	Phosphoinositide phospholipase C	PIPLC	-
L- Pyrrolydonyl- Arylamidase	PyrA	-	D-Xylose	dXYL	-
L-Arabitol	IARL	-	Arginine Dihydrolase1	ADH1	+
D-Cellobiose	dCEL	-	β–Galactosidase	BGAL	-
β–Galactosidase	BGAL	-	α –Glucosidase	AGLU	+
H_2_S production	H25	-	Ala-Phe-Pro-Arylamidase	APPA	-
β-N-Acetyl –Glucosaminidase	BNAG	-	Cyclodextrin	CDEX	+
Glutamyl Arylamidase pNA	AGLTp	+	L-Aspartate Arylamidase	AspA	+
D-Glucose	dGLU	+	β –Galactopyranosidase	BGAR	-
γ –Glutamyl –Transferase	GGT	+	α -Mannosidase	AMAN	-
Glucose Fermentation	OFF	-	Phosphatase	PHOS	-
β –Glucosidase	BGLU	-	Leucine Arylamidase	LeuA	-
D-Maltose	dMAL	-	L-Proline Arylamidase	ProA	-
D-Mannitol	dMAN	-	β –Glucuronidase	BGURr	-
D-Mannose	dMNE	+	α –Galactosidase	AGAL	-
β –Xylosidase	BXYL	-	L- Pyrrolydonyl- Arylamidase	PyrA	+
β -alanine arylamidase pNA	BAlap	+	β –Glucuronidase	BGUR	-
L-Proline Arylamidase	ProA	+	Alanine Arylamidase	AlaA	-
Lipase	LIP	+	Tyrosine Arylamidase	TyrA	-
Palatinose	PLE	-	D-Sorbitol	dSOR	+
Tyrosine Arylamidase	TyrA	-	Urease	URE	-
Urease	URE	-	Polymyxin B Resistance	POLYB	+
D-Sorbitol	dSOR	-	D-Galactose	dGAL	+
Saccharose/Sucrose	SAC	-	D-Ripose	dRIB	+
D-Tagatose	dTAG	-	L-Lactate Alkalinization	ILATk	-
D-Trehalose	dTRE	-	Lactose	LAC	+
Sodium Citrate	CIT	+	N-Acetyl-D-Glucosamine	NAG	+
Malonate	MNT	+	D-Maltose	dMAL	+
5-Keto-D-Gluconate	5KG	-	Bacitracin Resistance	BACI	+
L-Lactate alkalinization	ILATK	+	Novobiocin Resistance	NOVO	+
α –Glucosidase	AGLU	-	Growth in 6.5% NaCl	NC6.5	-
Succinate Alkalinization	SUCT	+	D-Mannitol	dMAN	+
β -N-Acetyl –Galactosaminidase	NAGA	-	D-Mannose	dMNE	+
α –Galactosidase	AGAL	-	Methyl-B-D-Glucopyranoside	MBdG	+
Phosphatase	PHOS	-	Pullulan	PUL	-
Glycine Arylamidase	GIyA	-	D-Raffinose	dRAF	-
Ornithine Decarboxylase	ODC	-	O/129 Resistance (Comp. Vibrio)	O129R	-
Lysine Decarboxylase	LDC	-	Salicin	SAL	+
L-Histidine Assimilation	IHISa	-	Saccharose/Sucrose	SAC	+
Courmarate	CMT	+	D-Trehalose	dTRE	+
β –Glucuronidase	BGUR	-	Arginine Dihydrolase2	ADH2s	+
O/129 Resistance (Comp. Vibrio)	O129R	+	Optochin Resistance	OPTO	+
Glu-Gly-Arg- Arylamidase	GGAA	-			
L-Malate Assimilation	IMLTa	+			
Ellman	ELLM	-			
L-Lactate Assimilation	ILATa	-			
**Probability**		**99%**			**99%**

α = Alpha, β = beta, γ = Gamma.

## Data Availability

Any other data will be available from the corresponding author upon request.

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
