# Peer review of "Enrofloxacin, Effective Treatment of Pseudomonas aeruginosa and Enterococcus faecalis Infection in Oreochromis niloticus"

_microorganisms, 2024, doi:10.3390/microorganisms12050901_

Round 1

Reviewer 1 Report (Previous Reviewer 1)

Comments and Suggestions for Authors

The authors cannot sufficiently revise the manuscript to address the concerns raised by the reviewers. 

Comments on the Quality of English Language

Moderate editing of English language required.

Author Response

Replay to Reviewer comments.

Dear respectful reviewers, thank you very much for the great effort in reviewing my manuscript, your guidance, valuable time and giving a chance to improve my manuscript, I Am confirming that all the indicated comments and corrections have been addressed as indicated, the corrections were highlighted in yellow color.

  • Reply to Reviewer 1 Comments:

Comment: The authors cannot sufficiently revise the manuscript to address the concerns raised by the reviewers.

Response: Dear respectful reviewer please specify and not give a general comment, Which of your previous comments were not handled by the authors? Despite this, the following is a second specific reply to your previous comments. 

  • Comment (1): Write solid findings in the abstract.

Response: All the solid findings of the study were mentioned in the abstract including:

  • Verification of aeruginosa and E. faecalis isolates.
  • Virulence test results (LD50) for Oreochromis niloticus.
  • Clinical signs and PM lesions appeared on the challenged fish.
  • Histopathological findings for challenged fish.
  • Antibiotic sensitivity results and MIC of enrofloxacin for aeruginosa and E. faecalis isolates.
  • Effective enrofloxacin dose for treatment of aeruginosa and E. faecalis infection.
  • Comment (2): The authors focused only Egypt, but this problem is also present globally.

 Response: Added in line 32 and 35 "Bacterial infections are the most abundant diseases affecting cultured fish [5]. Globally, fish diseases are estimated to contribute to more than 30% of the overall production loss, bacterial diseases represent a serious challenge for tilapia culture worldwide (6)"

  • Comment (3): Lines 54-55, I do not understand this section, why fish were obtained from a private farm? What was the purpose of it? Do you mean that you want to isolate bacteria from these fish or use for infectivity test?

Response: Fish mentioned in lines 58 to 62 (N=500) are the experimental fish that were used in the infectivity test (determining the LD50) and also were used in the treatment trial, we obtained these fish from a private farm as researchers normally get fishes for their experimental work from fish farms.

  • Comment (4): If you use these fish for the purpose of infectivity test, did you investigate these fish whether they are free from pathogens, or we can say that they are specific pathogen free?

Response: Thank you very much for remembering me. Ten randomly selected fish were dissected and inspected for the presence of parasitic infestation and microbiologically examined for the bacterial isolation. All of the examined fish were free, this section was added to materials and methods (Line 67-69).

  • Comment (5): Lines 75-76, how you confirmed aeruginosa and E. faecalis? What kind of tests were you used? Did you confirm them by sequencing?

Response: P. aeruginosa and E. faecalis were isolated and identified in a previous study and then preserved. Authors re-confirmed the identification of these isolates by using the selective differential media (Pseudomonas selective agar supplemented with cephalothin, Fucidin for P. aeruginosa and M-Enterococcus agar base media for E. faecalis). Isolates were further re-confirmed using VITEK 2 automatic biochemical bacterial identification system which is valid and considered the most accurate and reliable bacterial identification system. The characteristic biochemical profile of each microorganism was shown in table (1). 

  • Comment (6): The authors may check other antibiotics against aeruginosa and E. faecalis.

Response: Yes, authors used many antibiotics (data not published), but both isolates showed multiple antibiotic resistance profile for seven different antibiotics while both isolates were sensitive only to enrofloxacin and ceftiofur.

  • Comment (7): Please improve and change the whole discussion section.

Response: Was corrected as indicated in lines 293 to 296, 330 to 333 and 343 to 365.

  • Comment (8): Figures: the quality of all figures are very poor. They must supply high resolution figures. Additionally, scale bars are not provided in any of the figures.

Response: Thank you for your comment. The histopathological figures were corrected, regarding the scale bars, the magnification power is added in each figure legend.

  • Comment (9): Line 334, write solid conclusion of the study.

Response: Corrected in lines 368 to 372.

  • Comment (10): Tables 1 and 2: legends are missing.

Response: Titles of Tables 1 and 2 are present and so for all figure legends.

  • Comment (11): This manuscript requires professional English editing.

Response: Regarding the English language many sentences were corrected. This manuscript score was 8/10 in research square as shown in the attached PDF and score in Grammarly was 91% as in the attached report

Reviewer 2 Report (Previous Reviewer 2)

Comments and Suggestions for Authors

Thank you for your amendments to improve the study presentation and address the appropriate use of antibiotics to mitigate disease losses. The Discussion would benefit from the addition of citing references that mention the possible disadvantages (as well as the advantages) of the use of enrofloxacin to provide a balanced argument in the sustainable application of these agents and control measures (monitoring of antibiotic residues Line 351), testing isolates for antibiotic susceptibility to minimise antibiotic use as conducted in this study and need for stewardship and monitoring to mitigate these risks.

antibiotic residue monitoring

https://doi.org/10.1016/j.foodres.2020.109865

https://doi.org/10.1016/j.foodchem.2017.11.094

https://doi.org/10.1016%2Fj.heliyon.2024.e28193

stewardship

https://doi.org/10.1038/s41598-020-78849-3

https://doi.org/10.3390/antibiotics11111574

https://doi.org/10.1016/j.aquaculture.2021.736735

https://doi.org/10.1111/are.13967

AMR risks in clinical infection from use in animal production

https://doi.org/10.3390/antibiotics10101193

AMR environment

https://doi.org/10.3390/ph13080189

AMR control measures

https://doi.org/10.1016/j.envpol.2020.115854

Comments on the Quality of English Language

The content would benefit from language editing

Author Response

Replay to Reviewer comments.

Dear respectful reviewers, thank you very much for the great effort in reviewing my manuscript, your guidance, valuable time and giving a chance to improve my manuscript, I Am confirming that all the indicated comments and corrections have been addressed as indicated, the corrections were highlighted in yellow color.

  • Reply to Reviewer 2 Comments:

Comment: Thank you for your amendments to improve the study presentation and address the appropriate use of antibiotics to mitigate disease losses. The Discussion would benefit from the addition of citing references that mention the possible disadvantages (as well as the advantages) of the use of enrofloxacin to provide a balanced argument in the sustainable application of these agents and control measures (monitoring of antibiotic residues Line 351), testing isolates for antibiotic susceptibility to minimise antibiotic use as conducted in this study and need for stewardship and monitoring to mitigate these risks.

Response: Dear respectful reviewer, thank you very much for your comment, in response to your suggestion some relevant references were used in lines 353 to 358.

Reviewer 3 Report (New Reviewer)

Comments and Suggestions for Authors

It is very important to review the document, and to give order and coherence to what is intended to document, describe, and represent (it is recommended to improve the figures, especially those of histopathological sections that are clearly seen and the details that are intended to highlight are observed); In addition to taking care of the style, as well as some typing errors detected and that are marked in color on the document.

A thorough review of the entire document is recommended, because it has drafting errors (this can improve substantially both in substance and in form).

It is recommended to review and adapt some sections of the work, such as the discussion (in this section it is recommended to compare with studies that have been carried out in a similar way globally), considering the type of trial (antibiotics, bacterial agents and organisms used; And laboratory tests, taking care in these the correct form of the units of measurement expressed), and in the conclusions, these should be concise (only those obtained from the work).

Check the correct way of writing references; Both in each of the supporting paragraphs, and at the end of the list (following the guidelines and/or recommendations of the journal)

Comments on the Quality of English Language

On the writing were highlighted in color several aspects that the authors should consider for review and improvement (considering the style, the correct writing, and improve the writing)

Author Response

Replay to Reviewer comments.

Dear respectful reviewers, thank you very much for the great effort in reviewing my manuscript, your guidance, valuable time and giving a chance to improve my manuscript, I Am confirming that all the indicated comments and corrections have been addressed as indicated, the corrections were highlighted in yellow color.

  • Reply to Reviewer 3 Comments:
  • Comment (1): It is very important to review the document, and to give order and coherence to what is intended to document, describe, and represent (it is recommended to improve the figures, especially those of histopathological sections that are clearly seen and the details that are intended to highlight are observed); In addition to taking care of the style, as well as some typing errors detected and that are marked in color on the document.

Response: Dear respectful reviewer, thank you for your observation, in response to your comment figures 2, 3 & 4 were corrected, and the lesion markers was highlighted to be more obvious. All the typing errors were also corrected, and the entire manuscript was English edited using Grammarly software and get a score of 91% as represented in the attached pdf.

  • Comment (2): A thorough review of the entire document is recommended because it has drafting errors (this can improve substantially both in substance and in form).

Response: the present manuscript was reviewed for drafting errors as indicated by reviewer.

  • Comment (3): It is recommended to review and adapt some sections of the work, such as the discussion (in this section it is recommended to compare with studies that have been carried out in a similar way globally), considering the type of trial (antibiotics, bacterial agents and organisms used; And laboratory tests, taking care in these the correct form of the units of measurement expressed), and in the conclusions, these should be concise (only those obtained from the work)

Response: all the required corrections were performed in discussion lines (294 to 298) and lines (332 to 335).

  • Comment (5): Check the correct way of writing references; Both in each of the supporting paragraphs, and at the end of the list (following the guidelines and/or recommendations of the journal).

Response: references in the main text and at references section was checked and corrected according to the journal guidelines.

.

Round 2

Reviewer 1 Report (Previous Reviewer 1)

Comments and Suggestions for Authors

no

Reviewer 3 Report (New Reviewer)

Comments and Suggestions for Authors

It is important to recognize the work that authors have done

This manuscript is a resubmission of an earlier submission. The following is a list of the peer review reports and author responses from that submission.

Round 1

Reviewer 1 Report

Comments and Suggestions for Authors

This study identified the efficacy of enrofloxacin for the treatment of Pseudomonas aeruginosa and Enterococcus faecalis isolated from Oreochromis niloticus. Overall, this manuscript requires professional English editing. I personally suggest to the authors that they should discuss their experimental design and findings with the experts (fish microbiologists) before submitting the revised manuscript.

My major comments are here:

Abstract: Write solid findings in the abstract.

Introduction: The authors focused only Egypt but this problem is also present globally. Please improve the quality of the introduction and consider other studies.

Materials and methods:

Lines 54-55, I do not understand this section, why fish were obtained from a private farm? What was the purpose of it? Do you mean that you want to isolate bacteria from these fish or use for infectivity test?

If you use these fish for the purpose of infectivity test, did you investigate these fish whether they are free from pathogens or we can say that they are specific pathogen free?

Lines 75-76, how you confirmed P. aeruginosa and E. faecalis? What kind of tests were you used? Did you confirm them by sequencing?

The authors may check other antibiotics against P. aeruginosa and E. faecalis.

Discussion: please improve and change whole discussion section.

Figures: the quality of all figures are very poor. They must supply high resolution of figures. Additionally, scale bars are not provided in any of the figures.

Line 334, write solid conclusion of the study.

Tables 1 and 2: legends are missing.

Comments on the Quality of English Language

this manuscript requires professional English editing

Reviewer 2 Report

Comments and Suggestions for Authors

Aboyadak & Ali assessed the efficacy and dosage of enrofloxacin in treating bacterial infection caused by P. aeruginosa and E. faecalis in tilapia. Challenge studies estimated the LD50 for both species. The authors described in detail the external and internal symptoms of infection and histopathological changes. Antimicrobial susceptibility testing by disc diffusion and MIC indicated susceptibility to enrofloxacin for both species and a relatively lower susceptibility for P. aeruginosa compared to E. faecalis. Accordingly, twice the concentration of enrofloxacin was required in the medicated feed to protect against  P. aeruginosa infection.

The study was able to demonstrate that the application and estimate the appropriate dose of enrofloxacin needed within medicated feed that afforded protection against two bacterial pathogens in tilapia. However, these findings needs to be balanced by a discussion of the unwanted side effects of this antibiotic in the field due to antibiotic residue concerns and risk of antibiotic resistance dissemination. This has resulted in this antibiotic being banned in several aquaculture producing countries. The authors need to address this sustainability issue in the discussion if promoting the use of this antibiotic.

The content would benefit from the use of references throughout and provide full methods details and minor typos (highlighted).

Line 13 per fish? dots?

Line 21 italics

Line 34 Gram

Line 48 Insert brief justification for use of enrofloxacin given safety concerns

Are you proposing identification of the pathogen and MIC testing to minimise antibiotic use?

State attempts at best practice to reduce antibiotic exposure and risks

Line 67 burned?

Line 75 list strain numbers for both species

Line 80 add references for selective media

Line 85 monitored?

Line 88 35C seems high for fish isolates?

Line 94 did same OD give identical CFU ml-1 for both species? usually varies per species

Line 99 & 102 was the challenge dose confirmed by viable count?

Line 105 challenge?

Line 107 considered?

Line 128 supplier ENR disc, broths

Line 136 were the CLSI breakpoints valid at 35C and both species?

Line 143 199 mL? what was the final total volume of each tube, not clear

Line 150 temperature 37C different from disc diffusion 35C, check temperature CLSI valid for aquaculture isolates

Line 153 state control?

Line 158 i.p.?

Line 178 & 180 was control in range?

Line 179 Figure 1 zone diameters sizes stated do not match c & d plate zones, third zone size, not half

Line 191 & 192 showed

Line 213 light symbols would be clearer on Figures

Line 268 Table 3 states 10 6 inoculum, E. faecalis 2 x 10 7 Line 159

Line 319 were there any differences in symptoms and pathology between the two species that would aid differential diagnosis?

Line 329 Is enrofloxacin perfect (line 329)?

balance use of enrofloxacin with reported issues in aquaculture and bans in force in other countries using quinolones (reference throughout)

Line 332 insert reference justify use of enrofloxacin

Discussion points

Reproducibility of challenge studies and doses:

Explain how the mortality differed 67% & 54% Table 3 at 10 6

Line 158 P. aeruginosa 2 x 10 6 & E. faecalis 2 x 10 7 not 10 6

Previous challenge 42% vs 20% at 10 6 considerably lower

describe how your best practice and disease management approaches would reduce these risks associated with this antibiotic

Accurate diagnosis (Pseudomonas required double dose) etc, susceptibility testing, appropriate dose, length treatment, withdrawal period and removal antibiotic residues

Is antibiotic use sustainable in the long term, how can their use be reduced?

What are your recommendations for sustainable use of enrofloxacin going forward?

Comments on the Quality of English Language

Overall, the manuscript would benefit from minor editing to improve the writing style.